# Trends in Animal Shelter Management, Adoption, and Animal Death in Taiwan from 2012 to 2020

**DOI:** 10.3390/ani13091451

**Published:** 2023-04-24

**Authors:** Tzu-Yun Yan, Kendy Tzu-yun Teng

**Affiliations:** 1Department of Veterinary Medicine, National Taiwan University, Taipei City 106216, Taiwan; 2Department of Veterinary Medicine, College of Veterinary Medicine, National Chung Hsing University, Taichung City 40227, Taiwan

**Keywords:** animal shelter, animal welfare, Taiwan, euthanasia, stray animals

## Abstract

**Simple Summary:**

The issue of free-roaming and shelter animals is important to the Taiwanese public. Public animal shelters in Taiwan used to play an essential role in managing the number of free-roaming animals; this practice was banned in 2017. This current study investigated the trends in animal intakes and outcomes in public animal shelters and the workload of shelter veterinarians in Taiwan from 2012 to 2020. We found a decrease in the intakes and outcomes of shelter animals over time, although that trend reversed in 2018 with a slight increase in intakes. In replacement of euthanasia, adopting and returning animals to where they were captured after being neutered became the main reasons for animal outcomes. Counties with a higher human fertility rate were shown to have a higher adoption rate. Alarmingly, shelter veterinarians’ workload has increased from 2018 to 2020. Many veterinarians take care of more than 100 incoming and outgoing animals monthly, higher than what is regulated by the organisation laws based on the Animal Protection Act Article 14.3 in Taiwan. In this paper, we found decreased animal intakes and outcomes, shifted reasons for animal outcomes, and an increased veterinary workload in recent years. This information provides the foundation for future research on methods of improving shelter management and work conditions for shelter staff.

**Abstract:**

This current study investigated the trends in public animal shelter intakes and outcomes and the workload of shelter veterinarians in Taiwan from 2012 to 2020 and reports spatial, temporal, and socioeconomic factors associated with these trends. Information about the public animal shelter management of dogs and cats from all counties of Taiwan between 2012 and 2020 was acquired from the National Animal Shelter Management System of the Council of Agriculture Executive Yuan in Taiwan. Ridge regression followed by multivariable linear regression was conducted to examine the risk factors for animal intakes, outcomes, the number of adopted animals, and the estimated veterinary workload in public animal shelters. The intakes and outcomes of shelter animals significantly decreased over time. Euthanasia, which was performed in the shelters, was positively associated with shelter animal intakes and outcomes as it resulted in animal outcomes and thus facilitated the flow of animals in the shelters. Adoption and trap–neuter–vaccination–return, in replacement of euthanasia, became the main reasons for animal outcomes, and with every increase in human fertility rate, the monthly number of adopted animals over the number of animals entering shelters increased by 1.10% (95% CI: 0.21 to 2.00). The veterinary workload in the shelters of two counties exceeded what is regulated by law (i.e., 100 animals per veterinarian) in 2018 and increased to six counties in 2020. This current study reported important trends in the management of public animal shelters in Taiwan, the increasing workload of shelter veterinarians, and factors associated with these trends. It built the epidemiological foundation for future research on methods of improving shelter management and work conditions for shelter staff.

## 1. Introduction

Free-roaming dogs and cats (FDCs) are an important global One Health/One Welfare issue [1,2]. Approximately 75% of the 700 million dogs [3] worldwide and 30–80% of the cats in the United States [4] are free roaming. Many FDCs suffer from vehicle trauma, predation, diseases such as rabies and babesiosis, and hostile environments [3]. However, FDCs also have negative impacts on many wildlife species and humans [3,5]. Predation, disease, competition, disturbance, and hybridisation caused by free-roaming dogs have contributed to 11 vertebrate extinctions and potentially threaten 188 endangered species worldwide [6]. Cats are linked to the extinction of 63 species as a result of their predatory behaviour [7] and also host over 100 species of pathogens [8,9]. A wildlife rescue and research centre in Taiwan reported that the number of injured wildlife attacked by FDCs increased from 16 in 2011 to 54 in 2020 [10]. In Taiwan, a rescued pangolin was found for the first time to be infected with canine parvovirus-2 (CPV-2), whose strain was closely clustered with CPV-2c strains from domestic dogs [11]. FDCs also pose substantial zoonotic threats, including rabies, which kills approximately 60,000 people worldwide every year through dog bites [12,13]. *Toxoplasma gondii*, with cats as the final host, infects approximately 30% of the global human population [14]. For the reasons given above, the management of FDCs, including controlling population size, mitigating public health threats and negative impacts on wildlife, and improving FDCs’ health and welfare, is an important and urgent task.

Common approaches for managing the population of FDCs include trap–neuter–vaccinate–return (TNVR), sheltering, adoption, culling if not adopted, and a combination of these [3,15]. Among these methods, both intensive TNVR (with or without sheltering) and culling have been shown more effective for population control of FDCs than sheltering [3,16,17,18,19]. TNVR seems to outperform culling, as it increases the proportion of sterilised animals that compete with intact ones for local resources [19]. Furthermore, TNVR has been reported to improve the welfare of free-roaming dogs by decreasing fighting wounds and increasing their body condition scores [20,21]. Vaccination helps control rabies and reduces related long-term costs [19,22].

In Taiwan, the issue of FDCs and shelter animals has gained significant public interest. Several factors have contributed to the number of FDCs in Taiwan, including pet relinquishment by owners and breeders, free-roaming and lost pets, feeding FDCs by some members of the public, as well as low rates of pet registration and neutering pets and animals used for breeding [23]. Most animals being admitted to shelters in Taiwan are dogs, but the proportion varies depending on the shelter, and most are strays/owner-relinquished animals. In the past, whenever FDCs were reported by the public, they would be captured by animal control officers (ACOs) and sent to a public shelter, where shelter animals used to be euthanised if not adopted or reclaimed after 12 days of stay. This practice for population control has been banned in 2017 owing to the increase in public awareness about the ethics of killing shelter animals [24,25]. As a result, the number of euthanised animals has plummeted. However, new issues have also emerged [26,27]. An overload of shelter animals, in combination with limited housing capacity and limited financial and human resources in public animal shelters, has greatly undermined the welfare of shelter animals. Many shelter animals experience stress from overcrowding, exposure to new and unfamiliar animals, and noisy housing conditions [28,29]. Increased serum cortisol levels have been shown in shelter dogs compared to pet dogs within the first three days after they were captured and placed in animal shelters [30]. In some animal shelters with a poor environment, shelter cats are more likely to lose weight or have upper respiratory tract disease, which may potentially increase the risk of unassisted death [31,32]. Furthermore, the excessive number of animals in shelters should be considered not only an animal welfare issue but also a human welfare issue, as it increases the workload of shelter managers and veterinarians, who are already at high risk of burnout and other mental health issues [33,34,35]. Guidelines in Taiwan have been established that shelter managers and veterinarians should not be responsible for more than 40 and 100 shelter animals at any given time, respectively [36].

To effectively reduce the number of shelter animals, it is essential to identify factors that might have an impact on, or be associated with, the trends in shelter animal intakes and outcomes. Most studies focus on the traits of animals and human–animal relationships that are associated with relinquishment contributing to animal intakes [37,38] or successful adoption contributing to animal outcomes [39,40]. However, limited information is available in the literature on the association between socioeconomic factors and shelter animal intakes and outcomes. In Taiwan, a positive association was found between gross domestic product (GDP) and the number of animal intakes and outcomes [24]. An increase in human population density has been shown to be positively associated with the number of shelter animal intakes [41]. Urban lifestyles have been linked to behavioural problems in pets, such as aggression toward humans, destruction, and excessive barking [42,43], which has been linked to an increase in pet relinquishment in the United States [37,38]. In terms of shelter animal outcomes, shelter adoption rates are generally higher in urban versus rural areas [24], potentially resulting from more human resources that could be devoted to adequate behaviour training and veterinary treatment [44].

To better understand the current situation of animal flow in public shelters in Taiwan, this study aimed to investigate the trends in (a) overall and specific animal intakes and outcomes of public shelters in Taiwan from 2012 to 2020, (b) the monthly workload of shelter veterinarians from 2018 to 2020, and (c) factors associated with these trends. We hypothesized that the overall intakes and outcomes would decrease after the ban on euthanasia for population control as it could reduce the shelter capacity. We also speculated that metropolitan areas with more economic resources would have higher intake, outcome, and adoption rates. Last, we hypothesized that people without children might be more likely to have animals, potentially through adoption, resulting in a negative association between the human fertility rate and the adoption rate. The information generated from this current study is expected to help improve the management of public animal shelters and, eventually, have real, positive impacts on the welfare of animals and humans.

## 2. Materials and Methods

### 2.1. Study Population and Data Collection

All data were collected from the public domain, requiring no ethical approval.

Information about the public animal shelter management of dogs and cats from all counties of Taiwan between 2012 and 2020 was acquired from the National Animal Shelter Management System of the Council of Agriculture Executive Yuan in Taiwan [45] and verified by the manager of the system. Two shelter management datasets were used in this current study. Dataset 1 consisted of annual surveys of the number of animals entering (‘intakes’) and leaving (‘outcomes’) animal shelters by county between 2012 and 2020. The animal outcomes recorded include both those that were admitted in the same year, as well as those that were admitted in previous years but left the shelter in the current year. Dataset 1 also contained information about the reason for which an animal left the shelter, including (a) being adopted by an individual, a private shelter, or an animal protection organisation (‘adoption’), (b) being euthanised in the shelter (‘euthanasia’), or (c) death not through induced euthanasia (‘unassisted death’). Information about different ways of animal intakes was not available in the dataset.

Dataset 2 comprised monthly information on the number of shelter animal intakes and outcomes between 2018 and 2020. Dataset 2 also contained more detailed information about how animals entered or left the shelter than Dataset 1. The reasons for outcomes included animals (a) being reclaimed by the owners (‘reclamation’), (b) being adopted by individuals, private kennels, or animal protection organisations (‘adoption’), (c) being euthanised in the shelter (‘euthanasia’), (d) death due to disease or emaciation (‘unassisted death’), (e) being returned to where it was caught through a TNVR program (‘TNVR’), (f) escaping from the shelter or ‘other’. The reasons for intake included (a) FDCs being captured by ACOs (‘capture’), (b) FDCs being rescued by ACOs after being notified by citizens (‘rescue’), (c) FDCs being rescued by citizens (‘consignation’), (d) pets being relinquished by their owners (‘relinquishment’), (e) pets being confiscated by ACOs from their owners for violation of the Animal Protection Law [25] (‘confiscation’), and (f) other. Dataset 2 also included the data on the maximum shelter capacity for both dogs and cats in each county for 2020 (shown in Appendix A), calculated by dividing the available sheltering space by 5 m^2^ per animal [36].

Information about factors potentially associated with trends in the number of animal intakes and outcomes of public animal shelters in Taiwan from 2012 to 2020 was collected from various sources. From the National Statistics Bureau [46], we obtained information about (a) per capita gross domestic product by year from 2012 to 2020 (‘GDP’), (b) human population by county and year from 2012 to 2020 (‘population’), (c) percentage of people above 15 years old with a bachelor degree by county and year from 2012 to 2020 (‘higher education’), (d) ‘fertility rate’ (i.e., the birth population divided by the number of women between 15 to 49 years old then times 1000) by county and year from 2012 to 2020), (e) ‘population density’ over the area of each county by year from 2012 to 2020, (f) ‘plain area (10,000 m^2^) of a county’ by year from 2012 to 2020, and (g) ‘population density over the plain area’ for each county by year from 2012 to 2020. Information that might be relevant to animal flow in the shelters obtained from the Council of Agriculture Executive Yuan [47] included (a) the number of stray dogs in each county in 2018 (‘stray dogs’), (b) the number of pet dogs and cats by county in 2019 (‘total pets’), (c) the number of public animal shelter managers in each county in 2018 and 2020 (‘total managers’), (d) the number of ACOs in different counties in 2018 and 2020 (‘total ACOs’), and (e) the number of public animal shelter veterinarians in each county in 2018 and 2020 (‘total vets’). Each full-time manager, ACO, or veterinarian was counted as 1, and a part-time worker was counted as 0.5 in the following analyses. Monthly ‘working days’ were calculated for each month from 2018 to 2020. We also included ‘administrative divisions’ (i.e., metropolis, city, county, and outer island; the order indicates the level of urbanization) and ‘geographical divisions’ (i.e., north, south, west, east, and outer islands) of the counties in Taiwan [48]. Whether euthanasia for population control (‘policy of euthanasia for population control’) was banned by the Animal Protection Law was also considered.

### 2.2. Data Analysis

Data cleaning and management were performed using Microsoft Excel 2016 (Microsoft Corp., Redmond, WA, USA) and R version x64 4.1.1 [49] with RStudio interface [50], facilitated by R packages ‘tidyverse’ [51] and ‘dplyr’ [52]. Descriptive statistics, including plotting temporal and spatial trends in the number and reasons for shelter animal intakes and outcomes, were conducted with R packages ‘rgdal’ [53], ‘rgeos’ [54], ‘maptools’ [55], ‘sf’ [56], ‘gridExtra’ [57], ‘lubridate’ [58], ‘scales’ [59], and ‘ggpubr’ [60].

To explore the factors potentially associated with trends important to animal shelter management in Taiwan (elaborated below), indicators of the trends and their potential risk factors were identified as outcomes and covariates of linear models, respectively. Using data from Datasets 1 and 2, five indicators were generated for animal shelter management. They consisted of (a) monthly shelter animal intakes over the maximum shelter capacity, (b) monthly shelter animal outcomes over the maximum shelter capacity, (c) monthly number of adopted animals over the number of animals entering and staying at shelters, (d) yearly number of unassisted deaths over shelter animal intakes, and (e) monthly number of animals entering and leaving public animal shelters over the number of veterinarians. Dataset 1 was used to calculate indicators (a)–(d) from 2012 to 2020 (Table 1). For modelling animal shelter management from 2018 to 2020, indicators (a) to (c) and (e) were considered (Table 2). The formulas of indicators (c) for 2012 to 2020 and for 2018 to 2020 were slightly different due to the difference in the information between Dataset 1 and Dataset 2.

A univariable linear regression model was built for each indicator, with the covariates listed in Table 1 and Table 2, followed by a two-step multivariable linear regression. First, we transformed the data that violated the assumption of normality. After that, a Ridge regression model was built for each indicator using all covariates. Covariates with a coefficient larger than 0.01 were chosen for inclusion in the following model selection using the Akaike information criterion (AIC) and Bayesian Information Criterion (BIC). The best subset of variables from all possible combinations of the covariates (i.e., the final model) for each indicator was the one with the lowest weighted sum of standardised differences between AIC and BIC [61]. Normality and homoscedasticity were evaluated by visual inspection of the residual and residual-versus-fitted plots. The inferential analyses mentioned above were facilitated by the ‘tidymodels’ [62], ‘glmnet’ [63], ‘stats’ [49], ‘datasets’ [49], and ‘car’ [64] packages.

## 3. Results

### 3.1. Overall Shelter Flow

According to Dataset 1, from 2012 to 2017, the annual number of animal intakes and outcomes decreased substantially from 111,029 to 43,438 and from 87,082 to 40,275, respectively (Figure 1). After 2017, these numbers stabilised (intakes ranged from 39,626 to 48,164; outcomes ranged from 29,378 to 33,961). Between 2018 and 2020, dogs accounted for 89.3% (interquartile range: 79.0% to 96.5%) of the shelter animals.

The number of three-year averaged shelter animal intakes and outcomes at the county level were the largest in the metropolises, such as Taipei Metropolitan Area (including Taipei and New Taipei), Taoyuan, Taichung, Tainan, and Kaohsiung, and the smallest for the counties in the east (Figure 2), but most decreased substantially from 2012 to 2020. From 2012 to 2014, the 3-year average animal intakes were higher than the outcomes in 5 major metropolises, including New Taipei (the average number of shelter animal intakes: 13,564; the average number of shelter animal outcomes: 10,491), Taoyuan (9318; 8896), Taichung (9496; 8664), Tainan (11,472; 10,697), and Kaohsiung (9182; 6107), while the intakes and outcomes became similar for all counties in later years. These values are provided in Appendix A.

### 3.2. Specific Reasons for Animal Intakes and Outcomes

Using Dataset 1, the reasons for shelter animal outcomes changed markedly from 2012 to 2020 (Figure 3). More than 60% of the outgoing animals were euthanised in 2012, but the percentage has declined substantially since, falling to 1.89% in 2017 and almost 0 thereafter. Correspondingly, the proportion of animal outcomes due to adoption greatly increased from 36.38% to 91.06% during the same period overall in Taiwan. The percentage of animals that died unassisted decreased slightly over time, but this information was unavailable for 2012 and 2013.

Between 2018 and 2020, most animal intakes by public animal shelters resulted from either capture or rescue by animal control officers in most counties except Taitung County, where most animals (80.5%) were sent to the shelters by citizens (Figure 4). Relinquishment and confiscation of animals only accounted for a small percentage (0 to 7.1%) of animal intakes in most counties. Most shelter animal outcomes were the result of adoption, particularly in Taitung County (90.1%). TNVR also accounted for a large proportion of animal outcomes, especially in Taichung (n = 9603), Tainan (n = 9595), and Taoyuan (n = 5032). Although reclamation and unassisted death were relatively uncommon reasons for animal outcomes, the percentages of reclamation in Taipei (25.0%), Lienchiang County (20.8%), Keelung County (16.1%), and New Taipei (12.2%) were relatively high, and 39.1% of animal outcomes in Penghu County were due to unassisted death. The complete tables of the total number and percentage of animal intakes and outcomes at public animal shelters for different reasons in each county from 2018 to 2020 can be found in Appendix A.

### 3.3. Risk Factors for Overall and Specific Animal Intakes and Outcomes

#### 3.3.1. Univariable Linear Regression Results

The results of univariable linear regression for the number of animal intakes per month over the maximum shelter capacity from 2012 to 2020 and for all indicators are presented in Table 3 and Appendix A, respectively. Most of the examined factors seemed to have a significant association (i.e., with a *p*−value < 0.05) with animal intakes. From 2012 to 2020, public animal shelters in counties received animals with an average of 113.89 [95% confidence interval (CI): 97.66 to 130.13]% of the maximum shelter capacity, whereas the animal intake in cities (regression estimate: −68.58, 95% CI: −102.38 to −34.77), metropolises (regression estimate: −39.96, 95% CI: −66.48 to −13.44), and outer islands (regression estimate: −93.85, 95% CI: −133.63 to −54.07) did not exceed the maximum shelter capacity. Regarding other indicators, banning euthanising shelter animals was associated with a 17.49% (95% CI: 10.32 to 24.66) increase in the monthly adoption rate.

From 2018 to 2020, compared with ‘counties’, ‘metropolises’ had an increase of 21.58% (95% CI: 16.44 to 26.72) in shelter intakes and 21.03% (95% CI: 15.74 to 26.32) in shelter outcomes. In Taipei, 13.35% (95% CI: 10.82 to 15.88) of total shelter animals were adopted monthly on average, whereas the percentage in ‘Taitung County’ was 43.39% (95% CI: 39.82 to 46.97) higher than in ‘Taipei’.

#### 3.3.2. Multivariable Linear Regression Results

Final multivariable models for all indicators (i.e., model outcomes) are shown in Table 4, and detailed results of the final models can be found in Appendix A. Most models were well explained by the included covariates with an adjusted R^2^ ≥ 0.70, except for the models for the number of adopted animals over the shelter animal intakes per month and the number of unassisted dead animals over the shelter animal intakes per year. ‘Year’ and ‘county’ were in the final model of all indicators, and the interaction between ‘year’ and ‘county’ appeared in all models using information from Dataset 2. Although the models showed that animal intakes and outcomes decreased from 2012 to 2020, the trends were relatively stable when we focused only on 2018 to 2020. ‘Policy of euthanasia for population control’ was positively associated with shelter animal intakes and outcomes between 2012 and 2020 (i.e., the number of animal intakes and outcomes was larger before 2017), but its effect also depended on the county. With every increase in the ‘human fertility rate’, the monthly number of adopted animals over the shelter animal intakes increased by 1.10% (95% CI: 0.21 to 2.00).

### 3.4. Workload of Shelter Veterinarians

The workload of veterinarians varies in different years, months, and counties. Overall, the workload was the lowest in February during the year and in Chiayi County among all the counties, whose shelter veterinarians were responsible for 11.82 (95% CI: 6.96 to 20.29) animals in January 2018. Alarmingly, we found veterinary workload in the shelters of 2 counties exceeded what is regulated by law (i.e., 100 animals) in 2018 and increased to 6 counties in 2020. Particularly, Hsinchu County (from 119 to 195 animals), Pingtung County (from 35 to 201 animals), Changhua County (from 49 to 188 animals), and Hualien County (from 65 to 196 animals) had a sharp increase in the veterinary workload from 2018 to 2020.

## 4. Discussion

This current study focuses on the trends in public animal shelter intakes and outcomes and the workload of shelter veterinarians in Taiwan from 2012 to 2020 and reports factors that show an association. We used ridge regression combined with AIC and BIC to find the best subsets of covariates for each indicator, an advanced method that can prevent overfitting and is underused in companion animal epidemiology [65,66]. Although variations in animal intake, animal outcomes, adoption, unassisted death, and workload of veterinarians were detected in different counties over the years, the intakes and outcomes of shelter animals significantly decreased, and adoption and TNVR, in replacement of euthanasia, became the main reasons for animal outcomes. Alarmingly, we found that the workload of shelter veterinarians exceeded that regulated by law and increased from 2018 to 2020.

Over the years, it has been noticeable that the number of intakes in animal shelters has exceeded the number of outcomes. The gap between these 2 metrics was narrow between 2014 and 2017, as shelter animals were regularly euthanised after a 12-day stay. However, the largest gap was observed in 2012 and 2013, possibly due to a high number of unassisted deaths, for which data was unavailable in Dataset 1. Since 2018, the difference between intakes and outcomes has widened, indicating an increased length of stay for shelter animals and more animals being in care at the end of the year than at the start of the year. Shelter animal intake sharply decreased from 2012 to 2020, which was likely related to the decreasing euthanasia for population control, resulting in limited space for new animals in the shelters. To address this challenge, public shelters have been encouraged to work more closely with reputable private shelters while also investing in TNVR programs aimed at controlling the population of free-roaming dogs [67]. In addition, since 2015, the Animal Protection Law mandates dog and cat owners to register, microchip, and neuter their animals [25], which might help reduce the reproduction of FDCs and, thus, also reduce shelter intakes. Both subsidised and mandatory neutering of dogs and cats have been shown to reduce shelter intake [68,69]. However, an investigation report from the Legislature of Taiwan found that the percentages of dog and cat registrations and neutering among the registered dogs and cats were only 62% and 50%, respectively, in Taiwan in 2017 [70]. Metropolises were shown to be positively associated with shelter intakes as we hypothesised, which might be partly attributed to the lifestyles of metropolises. Behavioural problems in dogs, the top reason for relinquishment as reported by animal shelters [37,38], have been associated with urban areas [40,42,43]. Moreover, pet owners may be more likely to face rental-related issues in metropolises, which is another major reason for relinquishment [71]. However, in our results, the proportion of animal intake due to relinquishment did not seem to be higher in metropolises than in other administrative divisions. ‘Capture and rescue’ and/or ‘Consignations’ were the main reasons for animal intake in metropolises in Taiwan. Yet, in many cases, people pretend to consign a rescued animal but, in reality, relinquish it to the shelter [72].

The reasons for animal outcomes shifted dramatically during the study period. Unlike what was reported in a previous study, that euthanasia was the major strategy for reducing animals in shelters in Taiwan [24], we found an approximately 210-fold decrease in euthanasia over shelter outcomes from 2012 to 2020. Although the ban on euthanasia of healthy shelter animals since 2017 partly contributed to the reduction as we hypothesised, the euthanasia rate had already been decreasing every year between 2012 and 2016. It was possibly due to the decrease in euthanasia for population control resulting from the impact of a documentary named Twelve Nights (https://www.imdb.com/title/tt4141906/, accessed on 20 March 2023), launched in 2013. The movie documented the killings of shelter dogs after 12 days without being adopted or reclaimed. After the launch, public opinion against euthanasia for population control in Taiwan sparked, which eventually led to the ban on euthanasia for population control in 2017. In the univariable results, the adoption rate had a negative association with the policy of euthanasia for population control, which was excluded by the multivariable model. Although this association might be confounded by ‘year’, it could be linked to an increase in resources spent on the promotion of adoption when work for conducting euthanasia was reduced. Currently, adoption and TNVR are the main reasons for animal outcomes. These trends in shelter animal outcomes revealed by our study align with international trends. In Metro Denver in the United States, from 1989 to 2010, the proportion of euthanasia over all outcomes decreased by 77%, and the proportion of live release, including adoption, reclamation by owners, and transferring out, increased by 39% in 4 large Metro Denver animal shelters [73]. For all facilities licenced by The Pet Animal Care Facilities Act in Colorado from 2000 to 2015, euthanasia over animal intakes decreased by >50%, and adoption of animal intakes increased by 24.0% and 15.9% for dogs and cats, respectively [68].

We found a great discrepancy in the adoption rate among the different counties in Taiwan, which likely reflects the differences in local shelter management policies. In Taitung County, most (90.1%) shelter animal outcomes were the result of adoption from 2018 to 2020, which might be due to successful campaigning, public engagement, and behavioural training [74,75]. The fertility rate was found to be positively associated with adoption, which is different from what we hypothesised: people without children might be more likely to adopt animals. However, our results might only reflect that people in counties with higher fertility rates prefer to acquire an animal through adoption than by other means due to various reasons, such as adoption being cheaper than buying a pet.

The ratio of animals moving through the animal shelters relative to the number of shelter veterinarians in most counties increased from 2018 to 2020, especially in many of the rural counties, such as Hsinchu County and Hualien County, where each veterinarian cared for more than 100 incoming and outgoing animals monthly. This result revealed that there might be gaps in veterinarian capacity between metropolitan and counties/cities areas in Taiwan, requiring urgent action to secure the welfare of shelter animals and the mental health of shelter veterinarians in cities and counties. The demanding work overload, combined with a low salary (typical of veterinarians working in animal shelters compared to those in private clinics) and limited autonomy while working in the public sector, may all contribute to high occupational stress, which can further lead to burnout [76,77] and depression [78]. Additionally, the nature of work in animal shelters exposes shelter veterinarians to a range of adverse mental effects, such as compassion fatigue and post-traumatic stress [79,80]. These mental health issues, without proper support, may increase the risk of substance misuse and suicide [78,81] and result in a higher employee turnover rate and related costs, negatively impacting the care for animals [35]. Fawcett (2019) provided some solutions to this problem [76], such as offering work-related training, building a positive and efficient working environment, and providing an appropriate salary. Additionally, the introduction of social workers in animal shelters has also been proposed as a potential solution [82].

There are some limitations to this current study. Firstly, although our data were official, we could not verify the validity of the data or how the data were collected at each shelter. We were also informed that the numbers might be corrected anytime if errors were spotted. Secondly, the category of unassisted deaths was missing for 2012 and 2013. The number of animals that died unassisted was assigned to the category of euthanasia. Moreover, as information about the number of staff in animal shelters and the maximum shelter capacities was only available for 2018 and 2020, we assumed a linear trend in the number of staff during the period. In the calculation for veterinarian workload, we used shelter animal intakes and outcomes over the number of veterinarians to define the workload but did not consider the variation in the number of animals that were already in these shelters. Lastly, the results of this current study would be more valuable if the trends could have been divided into different animal species and ages.

## 5. Conclusions

Trends in the number of animals moving through the public animal sheltering system are an important welfare issue as they affect the welfare of animals, shelter staff, and even potential adopters. In this current study, we reported an overview of the situation of public animal shelters in Taiwan from 2012 to 2020 and found decreased animal intakes and outcomes, shifted reasons for animal outcomes, and an increased veterinary workload. Year and county are the most important factors associated with these trends when we used Ridge regression followed by multivariable linear regression. Counties with higher adoption rates, unassisted death rates, and veterinary workloads were identified. Based on these results, we will study how different shelter management strategies may affect the intakes, outcomes, and welfare of shelter animals in the future, as well as further investigate risk factors and potential solutions for the mental health issues of shelter staff.

## Figures and Tables

**Figure 1 animals-13-01451-f001:**
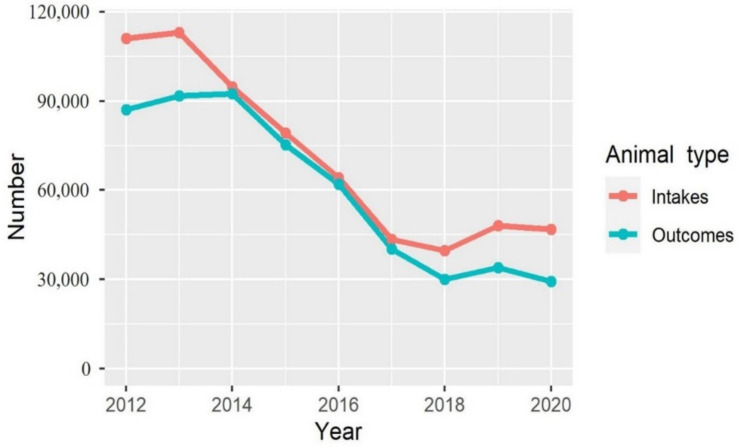
The number of annual public shelter animal intakes and outcomes from 2012 to 2020 in Taiwan.

**Figure 2 animals-13-01451-f002:**
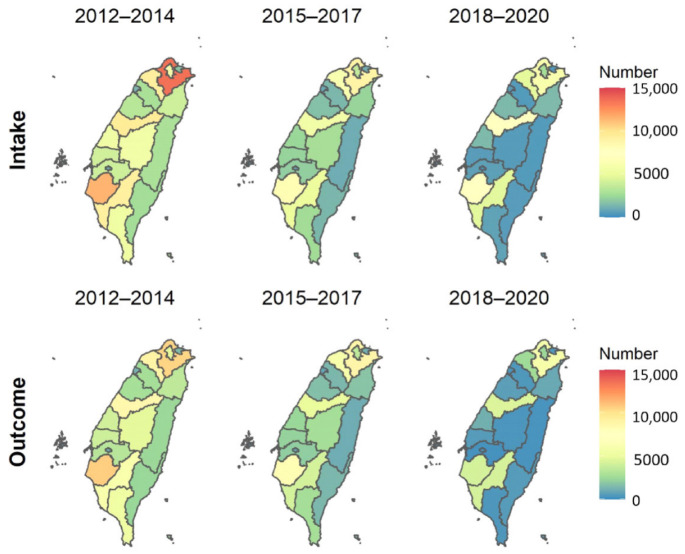
The three-year average shelter animal intakes and outcomes at the county level from 2012 to 2020 in Taiwan.

**Figure 3 animals-13-01451-f003:**
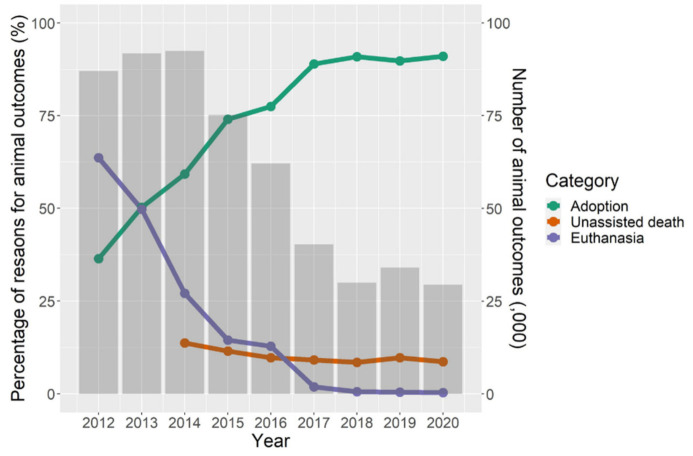
The number of outcomes and the percentage of different reasons for these outcomes at public animal shelters in Taiwan from 2012 to 2020.

**Figure 4 animals-13-01451-f004:**
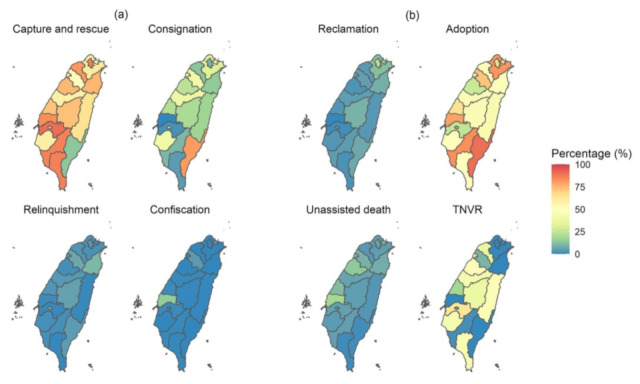
The percentage of different reasons for animal (**a**) intakes and (**b**) outcomes at the public animal shelters of different counties from 2018 to 2020. TNVR: trap–neuter–vaccinate–return.

**Table 1 animals-13-01451-t001:** Linear regression outcomes, equations for the calculation of the outcomes, and covariates for modelling the public animal shelter management in Taiwan from 2012 to 2020.

Indicators	Equations	Covariates
PI	$\left(\mathrm{AIY}÷12\right)÷SC\times 100$	(a) year, (b) county, (c) administrative division, (d) geographical divisions, (e) GDP, (f) population, (g) higher education, (h) plain area of a county, (i) population density over the plain area, (j) population density, (k) fertility rate, and (l) euthanasia
PO	$\left(\mathrm{AOY}÷12\right)÷SC\times 100$	
PA	AAY÷AIY×100	
PD	ADAY÷AIY×100	(a) year, (b) county, (c) administrative division, (d) geographical divisions, (e) higher education, and (f) euthanasia

PI: monthly shelter animal intakes over the maximum shelter capacity, PO: monthly shelter animal outcomes over the maximum shelter capacity, PA: the number of adopted animals over the shelter animal intakes per month, PD: the number of unassisted dead animals over the shelter animal intakes per year, AIY: the shelter animal intakes each year, SC: the maximum shelter capacity, AOY: the shelter animal outcomes each year, AAY: the number of adopted animals each year, ADAY: the number of animals that died in shelters each year.

**Table 2 animals-13-01451-t002:** Linear regression outcomes, equations for the calculation of the outcomes, and covariates for modelling the public animal shelter management in Taiwan from 2018 to 2020.

Indicators	Equations	Covariates
PI	AIM÷SC×100	(a) year, (b) month, (c) county, (d) administrative division, (e) geographical divisions, (f) GDP, (g) population, (h) higher education, (i) plain area of a county, (j) population density over the plain area, (k) population density, (l) fertility rate, (m) stray dogs, (n) total pets, (o) total managers, (p) total ACOs, (q) total vets, (r) working day
PO	AOM÷SC×100	
PA	$\mathrm{AAM}÷\left(\mathrm{AIM}+\mathrm{ABM}\right)\times 100$	
WV	$\left(\mathrm{AIM}+\mathrm{AOM}\right)÷NV$	(a) year, (b) month, (c) county, (d) administrative division, (e) geographical divisions

PI: monthly shelter animal intakes over the maximum shelter capacity, PO: monthly shelter animal outcomes over the maximum shelter capacity, PA: the number of adopted animals over the number of animals entering and staying at shelters per month, WV: the indicator for the monthly workload of shelter veterinarians, AIM: the shelter animal intakes each month, SC: the maximum shelter capacity for animals, AOM: the shelter animal outcomes each month, AAM: the number of adopted animals each month, ABM: the number of shelter animals at the beginning of the month, NV: the numbers of veterinarians working in public animal shelters in each county in 2021.

**Table 3 animals-13-01451-t003:** Univariable linear regression results for the number of shelter animal intakes per month over the maximum shelter capacity from 2012 to 2020.

Covariate	Category	Estimate (95% CI ^1^)	*p*-Value	*p*-Value for the Covariate
Year	Intercept	146.56 (127.02 to 166.09)	<0.001	<0.001
	Year	−15.70 (−19.81 to −11.60)	<0.001	
County	Intercept (Taipei)	56.38 (12.71 to 100.05)	0.012	<0.001
	Chiayi County	114.41 (52.65 to 176.17)	<0.001	
	Chiayi City	−3.69 (−65.45 to 58.07)	0.907	
	Changhua County	47.53 (−14.23 to 109.30)	0.133	
	Hsinchu County	49.37 (−12.40 to 111.13)	0.119	
	Hsinchu City	−37.07 (−98.83 to 24.70)	0.241	
	Hualien County	69.70 (7.94 to 131.46)	0.028	
	Kaohsiung	−0.71 (−62.47 to 61.05)	0.982	
	Keelung County	7.58 (−54.18 to 69.34)	0.810	
	Kinmen and Lienchiang County	−32.01 (−93.77 to 29.75)	0.311	
	Miaoli County	3.90 (−57.86 to 65.66)	0.902	
	Nantou County	10.20 (−51.56 to 71.96)	0.747	
	New Taipei	−16.68 (−78.44 to 45.08)	0.597	
	Penghu County	−40.66 (−102.42 to 21.11)	0.199	
	Pingtung County	211.52 (149.76 to 273.28)	<0.001	
	Taichung	69.70 (7.94 to 131.46)	0.028	
	Tainan	43.29 (−18.47 to 105.05)	0.171	
	Taitung County	51.43 (−10.33 to 113.20)	0.105	
	Taoyuan	9.74 (−52.02 to 71.51)	0.758	
	Yilan County	9.20 (−52.56 to 70.96)	0.771	
	Yunlin County	7.90 (−53.86 to 69.66)	0.802	
Administrative division ^2^	Intercept (County)	113.89 (97.66 to 130.13)	<0.001	<0.001
	City	−68.58 (−102.38 to −34.77)	<0.001	
	Metropolis	−39.96 (−66.48 to −13.44)	0.004	
	Outer island	−93.85 (−133.63 to −54.07)	<0.001	
Geographical division ^3^	Intercept (West)	84.22 (61.43 to 107.02)	<0.001	<0.001
	East	32.72 (−9.92 to 75.36)	0.134	
	North	−24.68 (−54.52 to 5.16)	0.107	
	Outer island	−64.18 (−106.82 to −21.54)	0.004	
	South	45.12 (12.89 to 77.35)	0.007	
GDP ^4^	Intercept	545.72 (423.40 to 668.04)	<0.001	<0.001
	Every 1000 New Taiwan Dollar	−0.63 (−0.79 to −0.46)	<0.001	
Population	Intercept	86.51 (69.21 to 103.81)	<0.001	0.662
	Every 100,000 people	−0.25 (−1.35 to 0.86)	0.662	
Higher education ^5^	Intercept	172.33 (132.46 to 212.20)	<0.001	<0.001
	%	−2.31 (−3.30 to −1.31)	<0.001	
Plain area of a county	Intercept	49.51 (31.91 to 67.11)	<0.001	<0.001
	10 km^2^	0.74 (0.45 to 1.03)	<0.001	
Population density over the plain area	Intercept	90.60 (76.88 to 104.33)	<0.001	0.048
	100,000 people/1 km^2^	−1.14 (−2.25E to −0.02)	0.048	
Population density	Intercept	96.23 (81.73 to 110.72)	<0.001	0.004
	Person/1 km^2^	−0.01 (−0.01 to −2.59E−03)	0.004	
Fertility rate	Intercept	144.94 (81.29 to 208.60)	<0.001	0.057
	‰	−1.90 (−3.84 to 0.04)	0.057	
Euthanasia ^6^	Intercept (No)	40.55 (24.44 to 56.66)	<0.001	<0.001
	Yes	77.75 (56.14 to 99.36)	<0.001	

^1^: confidence interval ^2^: administrative divisions in Taiwan (i.e., metropolis, city, county, and outer island) ^3^: geographical divisions (i.e., north, south, west, east, and outer island) ^4^: per capita gross domestic product ^5^: percentage of people above 15 years old with a bachelor’s degree by county and month ^6^: whether euthanasia for population control was enforced.

**Table 4 animals-13-01451-t004:** The covariates in the final multivariable linear regression models investigated the trends in animal intake/outcomes, adoption, unassisted death, and shelter veterinarian workload between 2012 and 2020 in Taiwan.

**Dataset 1 (2012–2020)**		
**Indicators**	**Covariates**	**Adjusted R^2^**
PI	(a) year (estimate: −0.08, 95% CI: −0.12 to −0.05)	0.88
	(b) county	
	(c) euthanasia (estimate: 0.19, 95% CI: −0.26 to 0.63)	
	(d) county: euthanasia	
PO	(a) year (estimate: −0.09, 95% CI: −0.11 to −0.06)	0.93
	(b) county	
	(c) euthanasia (estimate: 0.11, 95% CI: −0.25 to 0.47)	
	(d) county: euthanasia	
PA	(a) year (estimate: 5.11, 95% CI: 3.79 to 6.43)	0.47
	(b) county	
	(c) fertility rate (estimate: 1.10, 95% CI: 0.21 to 2.00)	
PD	(a) year (estimate: −0.06, 95% CI: −0.34 to 0.21)	0.66
	(b) county	
	(c) year: county	
**Dataset 2 (2018–2020)**		
**Indicators**	**Covariates**	**Adjusted R** ^2^
PI	(a) year (estimate: 0.04, 95% CI: −0.53 to 0.62)	0.76
	(b) county	
	(c) working day (estimate: 0.05, 95% CI: 0.04 to 0.06)	
	(d) year^2^ (estimate: −0.07, 95% CI: −0.34 to 0.21)	
	(e) year: county	
	(f) year^2^: county	
PO	(a) year (estimate: −0.06, 95% CI: −0.67 to 0.54)	0.76
	(b) county	
	(c) working day (estimate: 0.05, 95% CI: 0.03 to 0.06)	
	(d) year^2^ (estimate: −3.04E−03, 95% CI: −0.29 to 0.29)	
	(e) year: county	
	(f) year^2^: county	
PA	(a) year (estimate: −0.76, 95% CI: −1.77 to 0.26)	0.77
	(b) county	
	(c) working day (estimate: 0.03, 95% CI: 0.01 to 0.06)	
	(d) year^2^ (estimate: 0.15, 95% CI: −0.34 to 0.63)	
	(e) year: county	
	(f) year^2^: county	
WV	(a) year (estimate: −0.13, 95% CI: −0.29 to 0.03)	0.70
	(b) month	
	(c) county	
	(d) year: county	

PI: monthly shelter animal intakes over the maximum shelter capacity, PO: monthly shelter animal outcomes over the maximum shelter capacity, PA: the number of adopted animals over the number of animals entering and staying at shelters per month, PD: the number of unassisted death animals over the shelter animal intakes per year, WV: monthly workload of shelter veterinarians, CI: confidence interval, euthanasia: whether euthanasia for population control was enforced.

## Data Availability

The data used for the analyses in this current study can be found at: https://www.pet.gov.tw/AnimalApp/ReportAnimalsAcceptFront.aspx (accessed on 11 July 2021).

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
