# Peer review of "Trends in Animal Shelter Management, Adoption, and Animal Death in Taiwan from 2012 to 2020"

_animals, 2023, doi:10.3390/ani13091451_

Round 1

Reviewer 1 Report

First, I want to commend the authors for a well written paper on an important topic.  My comments are requests for clarification about topics in the paper and general editorial suggestions. 

Line 12- "We found a decrease in the intakes and outcomes of shelter animals over time, but the number increased in many counties between 2018 and 2020".  This sentence as written is confusing.  Maybe split into two sentences or consider rewording to suggest that the overall trend was a decrease although that trend reversed in 2018 with a slight increase in intakes and outcomes. 

Line 29-30- This sentence, as written, is very confusing.  How is euthanasia positively associated with shelter intakes and outcomes?  Maybe take this part of the sentence out of the abstract (leave it for the results where it can be explained) or split into two sentences where additional clarification can occur. 

Lines 48-50 is awkwardly written and is a run-on sentence.  Consider revising.

Lines 75-78- This is a run-on sentence.  Consider revising or split into two sentences for clarity.

Lines 105-114- This section belongs in results. 

Lines 108-111 conflicts with Lines 111-114.  Urban lifestyle = relinquishment and higher adoption rate in urban areas?

Lines 115-121- Why 2018-2020 for workload bur 2012-2018 for intake/outcomes?  How did you select these dates?

Line 243- Adoption greatly increased from 36.38% to 91.06%.  Overall or in specific counties?  

Line 252-255- Run on sentence.  Consider revising or split into 2 sentences.

Line 255-256- This information is out of place.  It needs to go with Line 243. 

Line 275- misplaced + sign?

Line 305-306- "The workload of every veterinarian increased by 0.92 each year from 2018-2020"   Everywhere or in certain counties?  This should have more info considering it was one of your specific aims.  Also, how do you justify this statement if you only have workload information from 2020?  

Line 318-320- This information should have been in results, not discussion.

Lines 328-330- "However, the percentages of dog and cat registrations and neutering were only 62% and 50% in 2017"  Please clarify this sentence.  Nationally or in specific regions?  How do you know 62% of all dogs were registered?  How was this information captured?  

Line 386-389- This limitation significantly impacts the information in your study.  Please consider revising your results based on the information provided here.  Indicated that a workload study for 2020 was conducted or something to that effect.  You cannot make assumptions this big with no data to support it.  

Author Response

Reviewer 1

First, I want to commend the authors for a well written paper on an important topic.  My comments are requests for clarification about topics in the paper and general editorial suggestions. 

Author response: We thank you for your time and comments on the manuscript. We believe that we have improved the manuscript by incorporating your suggestions.

Line 12- "We found a decrease in the intakes and outcomes of shelter animals over time, but the number increased in many counties between 2018 and 2020".  This sentence as written is confusing.  Maybe split into two sentences or consider rewording to suggest that the overall trend was a decrease although that trend reversed in 2018 with a slight increase in intakes and outcomes. 

Author response: We agree with you and have incorporated this suggestion in Lines 12-13.

We found a decrease in the intakes and outcomes of shelter animals over time although that trend reversed in 2018 with a slight increase in intakes.

Line 29-30- This sentence, as written, is very confusing.  How is euthanasia positively associated with shelter intakes and outcomes?  Maybe take this part of the sentence out of the abstract (leave it for the results where it can be explained) or split into two sentences where additional clarification can occur. 

Author response: We wrote like this because euthanasia for population control resulted in animal outcomes and thus facilitated the flow of animals in the shelters. We have elaborated on this in Lines 29-32 as shown below.  

The intakes and outcomes of shelter animals significantly decreased over time. Euthanasia, which was performed in the shelters, was positively associated with shelter animal intakes and outcomes as it resulted in animal outcomes and thus facilitated the flow of animals in the shelters. (Lines 29-32)

Lines 48-50 is awkwardly written and is a run-on sentence.  Consider revising.

Author response: We have revised the sentence as shown below:

Predation, disease, competition, disturbance, and hybridisation caused by free-roaming dogs have contributed to 11 vertebrate extinctions and potentially threaten 188 endangered species worldwide. (Line 50-52)

Lines 75-78- This is a run-on sentence.  Consider revising or split into two sentences for clarity.

Author response: We have revised as suggested, shown below.

In Taiwan, shelter animals used to be euthanised if not adopted after 12 days of stay. This has been banned in 2017 owing to the increase in public awareness about the ethics of killing shelter animals. (Lines 76-78)

Lines 105-114- This section belongs in results. 

Author response: Thank you for your suggestion. However, as this section is about the literature on the associations between socioeconomic factors and shelter intake and outcome, we think it would be more suitable in Introduction than in Results.

Lines 108-111 conflicts with Lines 111-114.  Urban lifestyle = relinquishment and higher adoption rate in urban areas?

Author response: Thank you for pointing this out. We now see that the words we chose had higher certainty than they should have. Thus, we have now made the sentences with greater uncertainty. Also, we agree that these two sentences seem conflicting at the first glance, but one factor (i.e., urban lifestyle) may have different types of relations with intakes and outcomes. Moreover, as literature on this topic is not abundant, the studies we cited were conducted in different countries. We have now acknowledged this. Please see the revised sentence as shown below.

Urban lifestyles have been linked to behavioural problems in pets, such as aggression toward humans, destruction, and excessive barking (Hsu et al., 2003; Puurunen et al., 2020), which has been linked to an increase in pet relinquishment in the United States (Salman et al., 2000; Segurson et al., 2005). In terms of shelter animal outcomes, shelter adoption rates are generally higher in urban versus rural areas (Peng et al., 2012), potentially resulting from more human resources that could be devoted to adequate behaviour training and veterinary treatment (Weiss et al., 2015). (Lines 109-115)

Lines 115-121- Why 2018-2020 for workload bur 2012-2018 for intake/outcomes?  How did you select these dates?

Author response: The survey on the number of staff in public animal shelters was not conducted every year in the past. We have phoned the Council of Agriculture in Taiwan and could only acquire the data in 2018 and 2020. Thus, we could only analyze the workload between 2018 and 2020. We also did the analysis for intake/outcomes using the more detailed information in 2018-2020 as shown in Table 2.

Line 243- Adoption greatly increased from 36.38% to 91.06%.  Overall or in specific counties?  

Author response: The result was in overall Taiwan. We have rewritten the sentence, shown as shown below.

Correspondingly, the number of animal outcomes due to adoption greatly increased from 36.38 % to 91.06 % during the same period overall in Taiwan. (Lines 252-253)

Line 252-255- Run on sentence.  Consider revising or split into 2 sentences.

Author response: Thank you for your suggestion. We have shortened and summarized the results in 3.2 and deleted the sentences.

Line 255-256- This information is out of place.  It needs to go with Line 243.

Author response: Thank you for your comment. In Line 243, we presented the overall trends in intakes and outcomes. Here in this paragraph, we showed the results of trends in intakes and outcomes of specific reasons, such as rescue, relinquishment, confiscation, adoption, TNVR, and death. Thus, we think the information should stay here.

Line 275- misplaced + sign?

Author response: Thank you for reminding us! We have deleted the ’+’.

Line 305-306- "The workload of every veterinarian increased by 0.92 each year from 2018-2020"   Everywhere or in certain counties?  This should have more info considering it was one of your specific aims.  Also, how do you justify this statement if you only have workload information from 2020?  

Author response: This is an important query. We have phoned the Council of Agriculture in Taiwan and acquired data on the number of animal shelter staff in 2018. We didn’t get the data in 2019 because the survey was not done every year. With the data in 2018 and 2020, we have redone the analysis with the assumption that the numbers linearly increased/decreased during the period between 2018 and 2020 in each county. With more complete data, we have supplemented more information and revised the results as shown below.

The workload of veterinarians varies in different years, months, and counties. Overall, the workload was the lowest in February during a year and in Chiayi County among all the counties, whose shelter veterinarians were responsible for 11.82 (95% CI: 6.96 to 20.29) animals in January 2018. Alarmingly, we found veterinary workload in the shelters of two counties exceeded what is regulated by law (i.e., 100 animals) in 2018 and increased to six counties in 2020. Particularly, Hsinchu County (from 119 to 195 animals), Pingtung County (from 35 to 201 animals), Changhua County (from 49 to 188 animals), and Hualien County (from 65 to 196 animals) had a sharp increase in the veterinary workload from 2018 to 2020. (Lines 310-318)

Line 318-320- This information should have been in results, not discussion.

Author response: We agree with you and move it to Rresults with some corrections. Please see as follows.

Alarmingly, we found veterinary workload in the shelters of two counties exceeded what is regulated by law (i.e., 100 animals) in 2018 and increased to six counties in 2020. Particularly, Hsinchu County (from 119 to 195 animals), Pingtung County (from 35 to 201 animals), Changhua County (from 49 to 188 animals), and Hualien County (from 65 to 196 animals) had a sharp increase in the veterinary workload from 2018 to 2020. (Lines 313-318)

Lines 328-330- "However, the percentages of dog and cat registrations and neutering were only 62% and 50% in 2017"  Please clarify this sentence.  Nationally or in specific regions?  How do you know 62% of all dogs were registered?  How was this information captured?  

Author response: The data is for the entire Taiwan, not specific regions, and we have now clarified this as shown below. We obtained this information from a report from the Legislature of Taiwan. We have phoned to inquire but could not obtain the information about the materials and methods.  

However, an investigation report from the Legislature of Taiwan found that the percentages of dog and cat registrations and neutering were only 62 % and 50 %, respectively, in Taiwan in 2017. (Lines 337-340)

Line 386-389- This limitation significantly impacts the information in your study.  Please consider revising your results based on the information provided here.  Indicated that a workload study for 2020 was conducted or something to that effect.  You cannot make assumptions this big with no data to support it.  

Author response: Thank you for your comment. We have now acquired data on the number of animal shelter staff in 2018. However, the survey was not conducted in 2019. Thus, we have redone the analysis with the assumption that the numbers linearly increased/decreased during the period between 2018 and 2020 in each county. With more complete data, we have supplemented more information and revised the results as shown below. We have now specified the process and the changes in results in the following lines.

The veterinary workload in the shelters of two counties exceeded what is regulated by law (i.e., 100 animals) in 2018 and increased to six counties in 2020. (Lines 35-36)

3.4. Workload of shelter veterinarians

The workload of veterinarians varies in different years, months, and counties. Overall, the workload was the lowest in February during a year and in Chiayi County among all the counties, whose shelter veterinarians were responsible for 11.82 (95% CI: 6.96 to 20.29) animals in January 2018. Alarmingly, we found veterinary workload in the shelters of two counties exceeded what is regulated by law (i.e., 100 animals) in 2018 and increased to six counties in 2020. Particularly, Hsinchu County (from 119 to 195 animals), Pingtung County (from 35 to 201 animals), Changhua County (from 49 to 188 animals), and Hualien County (from 65 to 196 animals) had a sharp increase in the veterinary workload from 2018 to 2020. (Lines 309-318)

Moreover, as information about the number of staff in animal shelters and the maximum shelter capacities was only available for 2018 and 2020, we assumed a linear trend in the number of staff during the period. (Lines 411-413)

Reviewer 2 Report

General comments:  This is an important topic, from a human- and animal-welfare perspective, and I’m pleased to see these data being analyzed and presented (and the problem of overloaded shelter veterinarians discussed).  As I understand it, the authors tracked the trends in the number of animals moving through animal shelters over time, using publicly-available datasets, and compared this to the number of veterinarians/cases per veterinarian over the same time period, with the goal of revealing a problem with the current system that needs addressing:  veterinarians are overloaded, even beyond the legal limit for cases per veterinarian, in many areas, and this may put the welfare of both the veterinarians, and the animals in their care, at risk. The amount of data the authors obtained and analyzed is impressive, and the analyses extensive. 

I did struggle a bit with a few aspects of the paper; I worry that, as written, the main message gets a bit lost in the volume of data presented within the manuscript text, and given some of the wording choices.  I’ve detailed these below:

T1) The use of the combined term “outcomes” was confusing, in that there is a big difference (from a welfare perspective) depending on the nature of the outcome – i.e., euthanasia vs. live adoption or TNVR.  This welfare risk could apply to both the animals as well as the staff who must perform euthanasia.  It was hard to evaluate many of the sheltering trends when data was presented as pooled ‘outcomes’, vs. data on # or proportion of specific outcomes.  If the point is to document the number of animals moving through the shelter system (animal “flow” perhaps) simply as a metric of staff workload, then (at least for this aspect of the study) I recommend focusing on a single metric representing the number of animals coming though the system (i.e., the number of animals these shelters much manage or ‘process’ – for lack of a better word – each year); or, more specifically, a single metric (ratio, proportion) showing number coming in vs. total space available/capacity for care.  If, on the other hand, the main goal was to describe trends in animal shelter adoption, death, etc. (as noted in the title), then I recommend breaking the pooled ‘outcomes’ measure down into live release vs euthanasia etc.  I suspect the paper is trying to accomplish both goals – these messages would be much clearer with some rearrangement of manuscript (particularly the methods/results) into two main subsections:  a) sheltering trends in terms of specific animal outcomes (before and after the 2017 law banning euthanasia of strays), and b) workload trends for shelter veterinarians and staff.

22)   I frequently felt like the important numbers and trends were getting lost in the sheer volume of data presented in the results section; I would recommend only including relevant data/results, and referring the reader to a Table or supplementary material for complete data. 

33) Along the same lines, it was difficult to follow the urban vs. rural (?) discussion – I assume the ‘counties’ mentioned were rural areas, vs. the metropolitan areas (and outlying islands?).  Many readers will not be familiar with the geography of Taiwan (although the message of the paper is certainly relevant beyond Taiwan) – I recommend either including a table early on clarifying with of the mentioned regions are rural vs. urban (based on population density?), or at least noting that the ‘counties’ are exclusively rural areas (?).  It would also be easier to understand if there was some direct comparison of trends in rural areas (as a whole) vs. metropolitan areas (as a whole); if this was in the paper, I missed it.

44) It might be helpful to state clear hypotheses in the intro (for example, with regard to trends in animal shelter populations relative to the 2017 law, trends in workloads for shelter veterinarians across the time period, or before/after the 2017 law…??), then discuss whether they were supported in the discussion (this might also help in reorganizing the results in a more understandable way?)  The conclusions surrounding the impact of the 2017 law (did it help, from an animal welfare perspective? Or did it cause problems indirectly, by overloading shelters, preventing some needy animals from being admitted, and shelter veterinarians, thereby potentially compromising quality of care? It is hard to tease this out from the paper as written.)

55)  It wasn’t entirely clear to me how the results of the linear regressions contributed to the conclusions.

Specific comments, with line #s:

10: suggest deleting ‘and’ and changing to animals; this practice

23: suggest add comma after 2020

29: see general comment about use of pooled ‘outcomes’ measure

29-30: confusing – why a positive relationship between euthanasia for population control and the number of animals moving through shelters?? I would think it would be the reverse (unless animals were taken into the shelters and euthanized there, in which case this should be clarified).  This kind of confusing point is also why I think it is so important to distinguish between specific outcomes for these animals.

32: is this human fertility rate?  Hard to understand why this would be related.  It is explained eventually in the discussion, but if included in the abstract/summary, I think it should be explained here (otherwise it distracts the reader from your main message)

34: ‘workload of shelter veterinarians exceeded that regulated by law’ – was this true for all years from 2018 to 2020? And why only these last three years? If linked to the 2017 law, I think that should be clarified here.

37: suggest replacing “them” with “these trends” for clarity

68: “better outcomes’ – in what sense? Do you mean, more effective means of population control? (it is hard to imagine that culling results in better outcomes for the dogs involved – I would avoid using ‘better’ to describe culling unless you clarify this

81-97: Nice description of the problem here!

110: suggest changing “resulting in” to “which can result in” or “which are common reasons for”

111-112: linked to shelters in urban vs. rural? Maybe reword to something like, “shelter adoption rates are generally higher in urban vs rural areas”??

112-113: awkward wording; maybe end sentence after (Peng et al., 2012). Then “Higher animal adoption rates have also been linked to adequate behavior training and veterinary care within the shelters, suggesting the importance of …”

131-134: apologies if I’ve missed it, but did you include data on the total # of available spaces for animals within the shelters (i.e., capacity of care of each facility/region being covered)? (and if so where did this data come from)

151: see previous comments about replacing pooled “outcomes” with something like, “the number of animals moving through the shelter system” (“outcomes” covers too many different results for the dogs, and it seems like the primary focus of the paper is the workload for shelter staff?)

156: why was human fertility rate chosen for inclusion in the models?

158-160:  what is ‘plain area’ and how does (e) differ from (g)?

162-163: does (a) refer to population size of stray dogs in the county? (and if so, how calculated?) or, number admitted to shelters?

166-167: was there data on the number or size (capacity for care, spaces available) of shelters by region (vs. # of managers, ACO’s – which wouldn’t necessarily accurately represent shelter capacity, esp. if staffing levels vary by region)

169-171: not sure this info is necessary (as administrative divisions etc. won’t shed light on the analyses for readers not familiar with Taiwan)

184: which trends specifically?

187-189 and 192-193: I would explain how the indicators were calculated when presented (otherwise confusing the reader reaches link 192-193) – even just starting line 186 sentence “Five indicators..” with “Using data from Datasets 1 and 2, five indicators were generated…” would be clearer. You could then explain which data went into which indicator later, as you have done (Lines 192-196).

203:  what does “outcome” mean here? (which outcomes, specifically)

218: see previous comments about “outcomes” – hard to understand trends in the different outcomes for animals (e.g., adoption vs euthanasia vs TNVR) when only a pooled outcome is analyzed and discussed in most sections

225-234: see general comment about detailed data listing in the manuscript; it would be clearer to summarize trends and then refer reader to Table for the data

225-234: see general comment about needing to clarify which regions are urban vs rural so that reader can follow your logic (different patterns in rural vs urban can be an important result, if you are aiming to improve shelter management practices country-wide)

239: clear definition of outcomes should be provided earlier in the paper (unless I missed it?); also, the information presented here (differences in trends in specific outcomes for animals) is very important (to readers, to the general public), but gets lost in the sea of numbers presented

242:  “in contrast”? or, “correspondingly”? as I would think a decrease in euthanasia as an outcome would go hand in hand with an increase in other (live) outcomes, such as adoption

Fig 3:  euthanasia rates were already drastically falling (and adoption rates increasing) before the 2017 change in laws about culling strays – how would you explain this?  I assume the drop in the number of animals moving through the system has to do with (as I believe you suggest later on) lack of available space, given that animals are no longer being rapidly euthanized in the shelter, thus freeing up space…?  If true, what is happening to these animals? (or, are the population rates of dogs/FRD’s decreasing overall during the same time period?)

249-264: I think much of these data/trends could be summarized (if relevant to the paper’s main goals; otherwise no need to summarize here) and the detailed results could be relocated to a table

275:  what does the + in +2012 refer to?

275-277:  these data (regarding public animal shelters in counties consistently receiving animals over the shelter capacity) are very important, I think; but, are the counties rural (as opposed to the city/metropolis etc mentioned on lines 278-279?  (see previous comment about the need to include characteristics of the different regions discussed)

281-282: banning euthanasia was associated with an increase in adoption rate – this in interesting; are possible reasons for this discussed in the Discussion?

282-285: not sure this information is necessary (seems tangential to the goals of the study)

288-292:  see previous comment about excess data (lists of numbers by county, etc.) included in manuscript body

292-295: potentially important results – can you present as a range of # animals veterinarians saw, and compare this to the legal limit? (or state as proportion of the legal limit?)

300:  re: ‘trends opposite in 2018-2020’ – this is potentially very interesting in light of the 2017 law, but doesn’t seem exactly line with Fig 1 (in which intakes begin to increase post 2017, but outcomes increase in one year but then drop back down)?

300-301:  re: ‘euthanasia was positively associated with shelter intakes and outcomes’ – I understand why more animals in could result in more animals euthanized (although, did this still hold true post 2017? It is hard to tell when you discuss the trend over the entire study period, including data from both before and after the law change); but if euthanasia is included in the overall ‘outcomes’ measure, then is it valid to conclude much from an association between increase in euthanasia and increase in outcomes?

303: add ‘human’ fertility rate (otherwise it sounds, in the circumstances, like you might be referring to canine fertility rate)

305: suggest listing the workload increase as a percentage, vs. a proportion

321-323: This statement is confusing as the ban on euthanasia happened in the middle of this trend (so, the factors influencing intake etc. are potentially very different pre-2017 vs post-2017).  And, the trend was not consistently downward, based on Figure 1 (intake decreased up until 2017, then began to increase; outcomes decreased then leveled off).  As written, this reads like an attempt to support the population culls, but not really supported by the data? (I may be confused here)

330-331: why metropolises? Is this simply a population density thing? (of humans? Or dogs? Or both?)

342: this is important (i.e., noting that the euthanasia rate had already been decreasing prior to 2017). – how do you explain this for Taiwan, and is it related in any way to the 2017 law? (i.e., does it reflect a changing attitude towards euthanasia that eventually led to the passing of the 2017 law?  Or, fewer dogs due to effective (?) population control?)

358-360: This is an interesting suggestion! (are there more expensive options widely available in the counties, e.g., professional dog breeders?)

362-363:  this might be clearer as a ratio? (something like, “The ratio of animals moving through the shelter relative to the number of shelter veterinarians increased from 2018-2020”) – although, not sure this is true of the ‘outcomes’ 2018-2020 (again based on Fig 1). 

363-364:  summarize for clarity, for readers unfamiliar with the region (so maybe, “especially in many of the rural counties, where each veterinarian…”)

368-369: suggest rewording as “Work overload, combined with a low salary (typical of veterinarians working in animal shelters compared to those in private clinics) and limited autonomy while working in the public sector, may all contribute to high occupational stress…”

384: I’m not sure I understand the significance (to the present study) of “the data could be updated at any time when needed”?

387-389: can you justify or defend this assumption? (vs. removing the years with no data from the analyses?) – these data seem important to your discussion of work overload for shelter vets

392-393: I’m not sure I would consider this a limitation of this study, as the 2017 law applied to both dogs and cats (correct?), and veterinary care is required for both

395: maybe reword to “Trends in number of animals moving through the animal sheltering system are an important welfare issue…”

397: add a comma after 2020

398: re: “shifted reasons for animal outcomes” – I think this is very important but it gets lost in the text, particularly given the pooled “outcomes” metric used throughout

Author Response

Reviewer 2

General comments:  This is an important topic, from a human- and animal-welfare perspective, and I’m pleased to see these data being analyzed and presented (and the problem of overloaded shelter veterinarians discussed).  As I understand it, the authors tracked the trends in the number of animals moving through animal shelters over time, using publicly-available datasets, and compared this to the number of veterinarians/cases per veterinarian over the same time period, with the goal of revealing a problem with the current system that needs addressing:  veterinarians are overloaded, even beyond the legal limit for cases per veterinarian, in many areas, and this may put the welfare of both the veterinarians, and the animals in their care, at risk. The amount of data the authors obtained and analyzed is impressive, and the analyses extensive. 

Author response: Thank you for your time and suggestion. We are very grateful for your comments on the manuscript. We have incorporated your advice and amended the relevant part of the manuscript.

I did struggle a bit with a few aspects of the paper; I worry that, as written, the main message gets a bit lost in the volume of data presented within the manuscript text, and given some of the wording choices.  I’ve detailed these below:

T1) The use of the combined term “outcomes” was confusing, in that there is a big difference (from a welfare perspective) depending on the nature of the outcome – i.e., euthanasia vs. live adoption or TNVR.  This welfare risk could apply to both the animals as well as the staff who must perform euthanasia.  It was hard to evaluate many of the sheltering trends when data was presented as pooled ‘outcomes’, vs. data on # or proportion of specific outcomes.  If the point is to document the number of animals moving through the shelter system (animal “flow” perhaps) simply as a metric of staff workload, then (at least for this aspect of the study) I recommend focusing on a single metric representing the number of animals coming though the system (i.e., the number of animals these shelters much manage or ‘process’ – for lack of a better word – each year); or, more specifically, a single metric (ratio, proportion) showing number coming in vs. total space available/capacity for care.  If, on the other hand, the main goal was to describe trends in animal shelter adoption, death, etc. (as noted in the title), then I recommend breaking the pooled ‘outcomes’ measure down into live release vs euthanasia etc.  I suspect the paper is trying to accomplish both goals – these messages would be much clearer with some rearrangement of manuscript (particularly the methods/results) into two main subsections:  a) sheltering trends in terms of specific animal outcomes (before and after the 2017 law banning euthanasia of strays), and b) workload trends for shelter veterinarians and staff.

Author response: Thank you for your comment. We have now made workload trends a separate section (3.4). As the reviewer suggested, we do want to report the overall outcomes and outcomes of specific reasons. The overall outcomes tell us about the level of flow of the animal shelters, and it can be used to estimate the workload as we did. We chose adoption and unassisted death for further modeling instead of categorizing all the outcomes into release vs euthanasia for two reasons. The first was that, as the reviewer mentioned, the nature of the outcome can be quite different. We thought combining adoption and TNVR together might tell us mixed information that could be difficult to untangle. We chose adoption and unassisted death for the model outcome because adoption may indicate the public concern for shelter animals and unassisted death may indicate the welfare states of animals in the shelters.

22)   I frequently felt like the important numbers and trends were getting lost in the sheer volume of data presented in the results section; I would recommend only including relevant data/results, and referring the reader to a Table or supplementary material for complete data. 

Author response: Thank you for your suggestions. We have restructured Result, shortened the text in section 3.2., and tried our best to only provide the most important information in the manuscript, especially in the sections of Abstract, Discussion, and Conclusions. We have to supply an extra table (i.e., Table 4) in the current revision as requested by another reviewer.

33) Along the same lines, it was difficult to follow the urban vs. rural (?) discussion – I assume the ‘counties’ mentioned were rural areas, vs. the metropolitan areas (and outlying islands?).  Many readers will not be familiar with the geography of Taiwan (although the message of the paper is certainly relevant beyond Taiwan) – I recommend either including a table early on clarifying with of the mentioned regions are rural vs. urban (based on population density?), or at least noting that the ‘counties’ are exclusively rural areas (?).  It would also be easier to understand if there was some direct comparison of trends in rural areas (as a whole) vs. metropolitan areas (as a whole); if this was in the paper, I missed it.

      Author response: Thank you for your suggestion. We have now used the administrative divisions when discussing our results, as shown below. Other “urban” and “rural” mentioned in our paper were extracted from the text of the papers that we cited.

      This result revealed that there might be gaps in veterinarian capacity between metropolitan and counties/cities areas in Taiwan, requiring urgent action to secure the welfare of shelter animals and the mental health of shelter veterinarians in cities and counties. (Lines 389-392)

44) It might be helpful to state clear hypotheses in the intro (for example, with regard to trends in animal shelter populations relative to the 2017 law, trends in workloads for shelter veterinarians across the time period, or before/after the 2017 law…??), then discuss whether they were supported in the discussion (this might also help in reorganizing the results in a more understandable way?)  The conclusions surrounding the impact of the 2017 law (did it help, from an animal welfare perspective? Or did it cause problems indirectly, by overloading shelters, preventing some needy animals from being admitted, and shelter veterinarians, thereby potentially compromising quality of care? It is hard to tease this out from the paper as written.)

      Author response: We have now supplemented the hypotheses as shown below. We have also discussed each of the hypotheses in Lines 120-125, 355-363, 340-342, and 380-381 respectively (shown below).

      …, and (c) factors associated with these trends. We hypothesized that the overall intakes and outcomes would decrease after the ban on euthanasia for population control as it could reduce the shelter capacity. We also speculated that metropolitan areas with more economic resources would have higher intake, outcome, and adoption rates. Last, we hypothesized that people without children might be more likely to have animals, potentially through adoption, resulting in a negative association between the human fertility rate and the adoption rate.  (Lines 120-125)

Although the ban on euthanasia of healthy shelter animals since 2017 partly contributed to the reduction as we hypothesised, the euthanasia rate had already been decreasing every year between 2012 and 2016. It was possibly due to the decrease in euthanasia for population control resulting from the impact of a documentary, named Twelve Nights, launched in 2013. The movie documented the killings of shelter dogs after 12 days without being adopted or reclaimed. After the launch, public opinion against euthanasia for population control in Taiwan sparked, which eventually led to the ban on euthanasia for population control in Taiwan in 2017. (Lines 355-363)

Metropolises were shown to be positively associated with shelter intakes as we hypothesised, which might be partly attributed to the lifestyles of metropolises. (Lines 340-342)

Fertility rate was found to be positively associated with adoption, which is different from what we hypothesised:… (Lines 380-381)

55)  It wasn’t entirely clear to me how the results of the linear regressions contributed to the conclusions.

      Author response: The result of linear regression models indicated risk factors for various indicators and suggested an increase in the veterinary workload. We have supplemented the conclusions as shown below.

      Year and county are the most important factors associated with these trends when we used Ridge regression followed by multivariable linear regression. (Lines 424-425)

Specific comments, with line #s:

10: suggest deleting ‘and’ and changing to animals; this practice

Author response: We have revised the text as suggested. (Line 10)

23: suggest add comma after 2020

Author response: We have revised the text as suggested. (Line 23)

29: see general comment about use of pooled ‘outcomes’ measure

Author response: Please see point T1) above.

29-30: confusing – why a positive relationship between euthanasia for population control and the number of animals moving through shelters?? I would think it would be the reverse (unless animals were taken into the shelters and euthanized there, in which case this should be clarified).  This kind of confusing point is also why I think it is so important to distinguish between specific outcomes for these animals.

Author response: It was the case, as you suggested, that many animals were euthanized for population control in animal shelters before 2017. As euthanasia was also performed for animals of poor welfare, we revised the sentence as follows:

The intakes and outcomes of shelter animals significantly decreased over time. Euthanasia, which was performed in the shelters, was positively associated with shelter animal intakes and outcomes as it resulted in animal outcomes and thus facilitated the flow of animals in the shelters. (Lines 29-32)

32: is this human fertility rate?  Hard to understand why this would be related.  It is explained eventually in the discussion, but if included in the abstract/summary, I think it should be explained here (otherwise it distracts the reader from your main message)

Author response: We hypothesized that people without children might be more likely to have animals or that people might choose a lifestyle of having pets than children, which would be reflected by fertility rate. We have incorporated your comments and supplemented Abstract and Introduction as shown below.

The current study investigated the trends in public animal shelter intakes and outcomes and the workload of shelter veterinarians in Taiwan from 2012 to 2020, and reports spatial, temporal, and socioeconomic factors associated with these trends. (Lines 22-24)

Adoption and trap-neuter-vaccination-return, in replacement of euthanasia, became the main reasons for animal outcomes, and with every increase in human fertility rate, the monthly number of adopted animals over the number of animals entering shelters increased by 1.10 % (95 % CI: 0.21 to 2.00). (Lines 32-35)

, and (c) factors associated with these trends. We hypothesized that the overall intakes and outcomes would decrease after the ban on euthanasia for population control as it could reduce the shelter capacity. We also speculated that metropolitan areas with more economic resources would have higher intake, outcome, and adoption rates. Last, we hypothesized that people without children might be more likely to have animals, potentially through adoption, resulting in a negative association between the human fertility rate and the adoption rate.  (Lines 120-125)

34: ‘workload of shelter veterinarians exceeded that regulated by law’ – was this true for all years from 2018 to 2020? And why only these last three years? If linked to the 2017 law, I think that should be clarified here.

Author response: We have now been more specific about the time and number of shelters that encountered the situation, as shown below:

The veterinary workload in the shelters of two counties exceeded what is regulated by law (i.e., 100 animals) in 2018 and increased to six counties in 2020. (Lines 35-36)

37: suggest replacing “them” with “these trends” for clarity

Author response: We have revised the text as suggested. (Line 38)

68: “better outcomes’ – in what sense? Do you mean, more effective means of population control? (it is hard to imagine that culling results in better outcomes for the dogs involved – I would avoid using ‘better’ to describe culling unless you clarify this

Author response: Thank you for your suggestion. We agree with you and have rewritten the text as shown below.

Among these methods, both intensive TNVR (with or without sheltering) and culling have been shown more effective for population control of FDCs than sheltering. (Lines 68-70)

81-97: Nice description of the problem here!

Author response: Thank you for your positive comment!

110: suggest changing “resulting in” to “which can result in” or “which are common reasons for”

Author response: We have revised the text as shown below:

…, which has been linked to an increase in pet relinquishment in the United States (Line 111)

111-112: linked to shelters in urban vs. rural? Maybe reword to something like, “shelter adoption rates are generally higher in urban vs rural areas”??

Author response: We agree with you and have rewritten the text as shown below.

In terms of shelter animal outcomes, shelter adoption rates are generally higher in urban versus rural areas (Peng et al., 2012), potentially resulting from more human resources that could be devoted to adequate behaviour training and veterinary treatment (Weiss et al., 2015). (Lines 112-115)

112-113: awkward wording; maybe end sentence after (Peng et al., 2012). Then “Higher animal adoption rates have also been linked to adequate behavior training and veterinary care within the shelters, suggesting the importance of …”

Author response: We agree with you. Please see point 111-112 above.

131-134: apologies if I’ve missed it, but did you include data on the total # of available spaces for animals within the shelters (i.e., capacity of care of each facility/region being covered)? (and if so where did this data come from)

Author response: Dataset 2 had the maximum shelter capacity for each county. We have added Lines 155-156 to clarify that.

Dataset 2 also included the data on the maximum shelter capacity in 2020 in each county. (Lines 155-156)

151: see previous comments about replacing pooled “outcomes” with something like, “the number of animals moving through the shelter system” (“outcomes” covers too many different results for the dogs, and it seems like the primary focus of the paper is the workload for shelter staff?)

Author response: As the reviewer suggested, we do want to report the overall outcomes and outcomes of specific reasons. The overall outcomes tell us about the level of flow of the animal shelters, and it can be used to estimate the workload as we did. We chose adoption and unassisted death for further modeling instead of categorizing all the outcomes into release vs euthanasia for two reasons. The first was that, as the reviewer mentioned, the nature of the outcome can be quite different. We thought combining adoption and TNVR together might tell us mixed information that could be difficult to untangle. We chose adoption and unassisted death for the model outcome because adoption may indicate the public concern for shelter animals and unassisted death may indicate the welfare states of animals in the shelters.

156: why was human fertility rate chosen for inclusion in the models?

Author response: Please see point 32 above.

158-160:  what is ‘plain area’ and how does (e) differ from (g)?

Author response: A previous paper in Lines 107-108 mentioned that FDCs often gathered in places with higher population densities. The terrain in Taiwan is complex, and some counties/cities have large mountain areas, which are habituated by fewer people than plain areas. Therefore, (g) the population density over the plain area might better reflect the true population density of a county/city. 

162-163: does (a) refer to population size of stray dogs in the county? (and if so, how calculated?) or, number admitted to shelters?

Author response: We obtained this information from the Council of Agriculture of Taiwan. The document reported that two-stage sampling to calculate the number of stray dogs in each county was used without providing further details.

166-167: was there data on the number or size (capacity for care, spaces available) of shelters by region (vs. # of managers, ACO’s – which wouldn’t necessarily accurately represent shelter capacity, esp. if staffing levels vary by region)

Author response: we incorporated the information of both the number of staff and the number of shelter animals in each county for the calculation of workload. We think this formula should reflect the workload of shelter staff.

169-171: not sure this info is necessary (as administrative divisions etc. won’t shed light on the analyses for readers not familiar with Taiwan)

Author response: Although it is true that most readers won’t be familiar with the administrative divisions in Taiwan, we think the administrative divisions are the indicator for economic resources. We wanted to explore the relationship between economic resources and shelter trends, and we think the information may be valuable to others.

184: which trends specifically?

Author response: We have tried to make it more clear here in Line 193, as shown follows:

To explore the factors potentially associated with trends important to animal shelter management in Taiwan (elaborated below), indicators of the trends and their potential risk factors were identified as outcomes and covariates of linear models, respectively. (Lines 192-194)

187-189 and 192-193: I would explain how the indicators were calculated when presented (otherwise confusing the reader reaches link 192-193) – even just starting line 186 sentence “Five indicators..” with “Using data from Datasets 1 and 2, five indicators were generated…” would be clearer. You could then explain which data went into which indicator later, as you have done (Lines 192-196).

Author response: Thank you for your suggestion and we have rewritten the text as shown below.

Using data from Datasets 1 and 2, five indicators were generated for animal shelter management. (Lines 194-196)

203:  what does “outcome” mean here? (which outcomes, specifically)

Author response: Thank you for reminding us. It means the five indicators and we have rewritten it as shown below.

A univariable linear regression model was built for each indicator, with the covariates listed in Tables 1 and 2, followed by a two-step multivariable linear regression. (Lines 212-213)

218: see previous comments about “outcomes” – hard to understand trends in the different outcomes for animals (e.g., adoption vs euthanasia vs TNVR) when only a pooled outcome is analyzed and discussed in most sections

Author response: As we replied in point T1) above, we do intend to report overall and specific outcomes. We also discussed the trends in specific outcomes in Lines 376-385.

225-234: see general comment about detailed data listing in the manuscript; it would be clearer to summarize trends and then refer reader to Table for the data

Author response: We have restructured Result, shortened the text in section 3.2., and tried our best to only provide the most important information in the manuscript, especially in the sections of Abstract, Discussion, and Conclusions. We have to supply an extra table (i.e., Table 4) in the current revision as requested by another reviewer.

225-234: see general comment about needing to clarify which regions are urban vs rural so that reader can follow your logic (different patterns in rural vs urban can be an important result, if you are aiming to improve shelter management practices country-wide)

Author response: Thank you for your suggestion. We have now used the administrative divisions when discussing our results to make sure the information is applicable to other countries and regions. Other “urban” and “rural” mentioned in our paper were extracted from the text of the papers that we cited.

239: clear definition of outcomes should be provided earlier in the paper (unless I missed it?); also, the information presented here (differences in trends in specific outcomes for animals) is very important (to readers, to the general public), but gets lost in the sea of numbers presented

Author response: We have now added the definitions in Lines 135-137, shown below. We used Dataset 1 here to obtain this result, and, as mentioned in Lines 249-250, the outcomes of Dataset 1 were only divided into three types: euthanasia, adoption, and unassisted death. We have clarified the text as shown below. 

Dataset 1 consisted of annual surveys of the number of animals entering (‘intakes’) and leaving (‘outcomes’) animal shelters by county between 2012 and 2020. (Lines 135-137)

Using Dataset 1, the reasons for shelter animal outcomes changed markedly from 2012 to 2020 (Figure 3). (Lines 249-250)

242:  “in contrast”? or, “correspondingly”? as I would think a decrease in euthanasia as an outcome would go hand in hand with an increase in other (live) outcomes, such as adoption

Author response: We agree with you and we have rewritten the text in Lines 252-253.

Correspondingly, the number of animal outcomes due to adoption greatly increased from 36.38 % to 91.06 % during the same period overall in Taiwan. (Lines 252-253)

Fig 3:  euthanasia rates were already drastically falling (and adoption rates increasing) before the 2017 change in laws about culling strays – how would you explain this?  I assume the drop in the number of animals moving through the system has to do with (as I believe you suggest later on) lack of available space, given that animals are no longer being rapidly euthanized in the shelter, thus freeing up space…?  If true, what is happening to these animals? (or, are the population rates of dogs/FRD’s decreasing overall during the same time period?)

Author response: Thank you for pointing this out. We think it was still due to the decrease in euthanasia for population control, and it was from the impact of a documentary, named Twelve Nights, launched in 2013. The movie documented the killings of shelter dogs after 12 days without being adopted or reclaimed. After the launch, public opinion against euthanasia for population control in Taiwan sparked, which eventually led to the ban on euthanasia for population control in Taiwan. A survey of the number of FRDs from the Council of Agriculture in 2009, 2015, 2018, and 2020 suggested an increase in the number, from 84,891 to 155,869 in Taiwan. We have rewritten Lines 331-333 and Lines 355-363 to clarify, shown as follows:

Shelter animal intake sharply decreased from 2012 to 2020, which was likely related to the decreasing euthanasia for population control, resulting in limited space for new animals in the shelters. (Lines 331-333)

Although the ban on euthanasia of healthy shelter animals since 2017 partly contributed to the reduction as we hypothesised, the euthanasia rate had already been decreasing every year between 2012 and 2016. It was possibly due to the decrease in euthanasia for population control resulting from the impact of a documentary, named Twelve Nights, launched in 2013. The movie documented the killings of shelter dogs after 12 days without being adopted or reclaimed. After the launch, public opinion against euthanasia for population control in Taiwan sparked, which eventually led to the ban on euthanasia for population control in Taiwan in 2017. (Lines 355-363)

249-264: I think much of these data/trends could be summarized (if relevant to the paper’s main goals; otherwise no need to summarize here) and the detailed results could be relocated to a table

Author response: We agree with you. We have trimmed and summarized the results 3.2 as shown below.

Between 2018 and 2020, most animal intakes by public animal shelters resulted from either capture or rescue by animal control officers in most counties except Taitung County, where most animals (80.5 %) were sent to the shelters by citizens (Figure 4). Relinquishment and confiscation of animals only accounted for a small percentage (0 to 7.1 %) of animal intakes in most counties. Most shelter animal outcomes were the result of adoption, particularly in Taitung County (90.1 %). TNVR also accounted for a large proportion of animal outcomes, especially in Taichung (n = 9,603), Tainan (n = 9,595), and Taoyuan (n = 5,032). Reclamation and unassisted death were relatively uncommon reasons for animal outcomes. The complete tables of the total number and percentage of animal intakes and outcomes at public animal shelters for different reasons in each county from 2018 to 2020 can be found in Table S3. (Lines 259-269)

275:  what does the + in +2012 refer to?

Author response: Thank you for reminding me! The ‘+’ is a mistake and we have deleted it.

275-277:  these data (regarding public animal shelters in counties consistently receiving animals over the shelter capacity) are very important, I think; but, are the counties rural (as opposed to the city/metropolis etc mentioned on lines 278-279?  (see previous comment about the need to include characteristics of the different regions discussed)

Author response: Yes, counties are more rural than cities and metropolitan areas. We have now indicated this in Lines 176-177, shown below:

We also included ‘administrative divisions’ (i.e., metropolis, city, county, and outer island; the order indicates the level of urbanization) and…

281-282: banning euthanasia was associated with an increase in adoption rate – this in interesting; are possible reasons for this discussed in the Discussion?

Author response: Although the association was detected in the univariable model, it was not as strong as other factors and excluded by the multivariable model. This association might be confounded by ‘year’. However, as work for conducting euthanasia decreased, more resources might be used to promote adoption, contributing to this result. We have added this in Discussion as shown below:

, which eventually led to the ban on euthanasia for population control in Taiwan in 2017. In the univariable results, adoption rate had a negative association with the policy of euthanasia for population control, which was excluded by the multivariable model. Although this association might be confounded by ‘year’, it could be linked to an increase in resources spent on the promotion of adoption when work for conducting euthanasia was reduced. (Lines 362-367)

282-285: not sure this information is necessary (seems tangential to the goals of the study)

Author response: We have removed the information here.

288-292:  see previous comment about excess data (lists of numbers by county, etc.) included in manuscript body

Author response: As we only supplied the univariable model results that were done using Dataset 2, we wanted to provide some results here. We have shortened the sentence, as shown below:

From 2018 to 2020, compared with ‘counties’, ‘metropolises’ had an increase of 21.58 % (95 % CI: 16.44 to 26.72) in shelter intakes and 21.03 % (95 % CI: 15.74 to 26.32) in shelter outcomes. In Taipei, 13.35 % (95 % CI: 10.82 to 15.88) of total shelter animals were adopted monthly on average, whereas the percentage in ‘Taitung County’ was 43.39 % (95 % CI: 39.82 to 46.97) higher than in ‘Taipei’. (Lines 288-292)

292-295: potentially important results – can you present as a range of # animals veterinarians saw, and compare this to the legal limit? (or state as proportion of the legal limit?)

Author response: We have removed this as it was sort of repeated. Please see Lines 309-318 (as shown below), which also included the comparison between the results and the legal limit.

300:  re: ‘trends opposite in 2018-2020’ – this is potentially very interesting in light of the 2017 law, but doesn’t seem exactly line with Fig 1 (in which intakes begin to increase post 2017, but outcomes increase in one year but then drop back down)?

Author response: We agree with you that outcomes became relatively stable instead of gradually increasing. We have clarified it as shown below.

Although the models showed that animal intakes and outcomes decreased from 2012 to 2020, the trends were relatively stable when we focused only on 2018 to 2020. (Lines 295-297)

300-301:  re: ‘euthanasia was positively associated with shelter intakes and outcomes’ – I understand why more animals in could result in more animals euthanized (although, did this still hold true post 2017? It is hard to tell when you discuss the trend over the entire study period, including data from both before and after the law change); but if euthanasia is included in the overall ‘outcomes’ measure, then is it valid to conclude much from an association between increase in euthanasia and increase in outcomes?

Author response: We understand the concern. Here ‘euthanasia’ was referred to as the policy of ‘euthanasia for population control’. We are sorry for the confusion. As a policy of euthanasia for population control does not necessarily contribute to the number of euthanasia, we think it is valid to have this factor in the model. We have modified the sentence as:

Policy of euthanasia for population control’ was positively associated with shelter animal intakes and outcomes between 2012 and 2020 (Lines 297-299)

303: add ‘human’ fertility rate (otherwise it sounds, in the circumstances, like you might be referring to canine fertility rate)

Author response: We have supplemented the sentences as shown below.

With every increase in the human fertility rate’, the monthly number of adopted animals over the shelter animal intakes increased by 1.10 (95 % CI: 0.21 to 2.00) %. (Lines 300-302)

305: suggest listing the workload increase as a percentage, vs. a proportion

Author response: We have removed this part.

321-323: This statement is confusing as the ban on euthanasia happened in the middle of this trend (so, the factors influencing intake etc. are potentially very different pre-2017 vs post-2017).  And, the trend was not consistently downward, based on Figure 1 (intake decreased up until 2017, then began to increase; outcomes decreased then leveled off).  As written, this reads like an attempt to support the population culls, but not really supported by the data? (I may be confused here)

Author response: We have changed this to:

Shelter animal intake sharply decreased from 2012 to 2020, which was likely related to the decreasing euthanasia for population control, resulting in limited space for new animals in the shelters. (Lines 331-333).

330-331: why metropolises? Is this simply a population density thing? (of humans? Or dogs? Or both?)

Author response: Thanks for pointing this out. We have given more thoughts, which can be found in the following sentences:

Moreover, pet owners may be more likely to face rental-related issues in metropolises, which is another major reason for relinquishment (Weiss et al., 2014). However, in our results, the proportion of animal intake due to relinquishment did not seem to be higher in metropolises than in other administrative divisions. ‘Capture and rescue’ and/or ‘Consignations’ were the main reasons for animal intake in metropolises in Taiwan. Yet, in many cases, people pretend to consign a rescued animal but, in reality, relinquish it to the shelter. (Lines 345-351)

342: this is important (i.e., noting that the euthanasia rate had already been decreasing prior to 2017). – how do you explain this for Taiwan, and is it related in any way to the 2017 law? (i.e., does it reflect a changing attitude towards euthanasia that eventually led to the passing of the 2017 law?  Or, fewer dogs due to effective (?) population control?)

Author response: Yes, we have the same postulation as yours. As in the response to Fig3, we think it was still due to the decrease in euthanasia for population control, and it was from the impact of a documentary, named Twelve Nights, launched in 2013. The movie documented the killings of shelter dogs after 12 days without being adopted or reclaimed. After the launch, public opinion against euthanasia for population control in Taiwan sparked, which eventually led to the ban on euthanasia for population control in Taiwan. We have rewritten Lines 355-363 to clarify.

358-360: This is an interesting suggestion! (are there more expensive options widely available in the counties, e.g., professional dog breeders?)

Author response: The price of a dog or a cat in the pet shop ranged from 10,000 to 50,000 New Taiwanese Dollars in Taiwan, not to mention the price at professional animal breeders. In contrast, people can adopt a pet from animal shelters without any cost. We have clarified the sentences as shown below.

However, our results might only reflect that people in counties with higher fertility rates prefer to acquire an animal through adoption than by other means due to various reasons, such as adoption being cheaper than buying a pet. (Lines 382-385)

362-363:  this might be clearer as a ratio? (something like, “The ratio of animals moving through the shelter relative to the number of shelter veterinarians increased from 2018-2020”) – although, not sure this is true of the ‘outcomes’ 2018-2020 (again based on Fig 1). 

Author response: We have rewritten the sentences as shown below:

The ratio of animals moving through the animal shelters relative to the number of shelter veterinarians in most counties increased from 2018 to 2020,… (Lines 386-387)

363-364:  summarize for clarity, for readers unfamiliar with the region (so maybe, “especially in many of the rural counties, where each veterinarian…”)

Author response: We have clarified the sentences as shown below.

…especially in many of the rural counties, such as Hsinchu County and Hualien County, where each veterinarian cared for more than 100 incoming and outgoing animals monthly. (Lines 387-389)

368-369: suggest rewording as “Work overload, combined with a low salary (typical of veterinarians working in animal shelters compared to those in private clinics) and limited autonomy while working in the public sector, may all contribute to high occupational stress…”

Author response: We have rewritten the sentences as shown below.

Work overload, combined with a low salary (typical of veterinarians working in animal shelters compared to those in private clinics) and limited autonomy while working in the public sector, may all contribute to high occupational stress and further lead to burnout. (Lines 392-395)

384: I’m not sure I understand the significance (to the present study) of “the data could be updated at any time when needed”?

Author response: We were informed that the numbers might be corrected anytime if errors were spotted. We have revised the sentence to, hopefully, make it clear:  

We were also informed that the numbers might be corrected anytime if errors were spotted.  (Lines 408-409)

387-389: can you justify or defend this assumption? (vs. removing the years with no data from the analyses?) – these data seem important to your discussion of work overload for shelter vets

Author response: As responded above, we have now acquired data on the number of animal shelter staff in 2018 and redone the analysis with the assumption that the numbers linearly increased/decreased during the period between 2018 and 2020 in each county. The sentence has been modified as:

Moreover, as information about the number of staff in animal shelters and the maximum shelter capacities was only available for 2018 and 2020, we assumed a linear trend in the number of staff during the period. (Lines 411-413)

392-393: I’m not sure I would consider this a limitation of this study, as the 2017 law applied to both dogs and cats (correct?), and veterinary care is required for both

Author response: We agree that vet care is required for both, but the amount of care that a dog and a cat need may be different. Also, we would much prefer to see the trends in specific intakes and outcomes by species than them being pooled together.

395: maybe reword to “Trends in number of animals moving through the animal sheltering system are an important welfare issue…”

Author response: Thank you for your suggestion and we have written the sentences as shown below.

Trends in the number of animals moving through the public animal sheltering system are an important welfare issue as they affect the welfare of animals, shelter staff, and even potential adopters. (Lines 419-421)

397: add a comma after 2020

Author response: Thank you for your suggestion. We have added it.

398: re: “shifted reasons for animal outcomes” – I think this is very important but it gets lost in the text, particularly given the pooled “outcomes” metric used throughout

Author response: We have tried to address this issue by emphasizing the importance of understanding the potential changes of the reasons for animal intakes and outcomes during the years throughout the manuscript. Some examples are provided below:

In this paper, we found decreased animal intakes and outcomes, shifted reasons for animal outcomes, and an increased veterinary workload in recent years. (Lines 18-19)

To better understand the current situation of animal flow in public shelters in Taiwan, this study aimed to investigate the trends in (a) overall and specific animal intakes and outcomes of public shelters in Taiwan from 2012 to 2020, (b)… (Lines 116-118)

Although variations in animal intake, animal outcomes, adoption, unassisted death, and workload of veterinarians were detected in different counties over the years, the intakes and outcomes of shelter animals significantly decreased, and adoption and TNVR, in replacement of euthanasia, became the main reasons for animal outcomes. (Lines 325-328)

Reviewer 3 Report

Authors aims to investigate the trends in (a) animal intakes and outcomes of public shelters in Taiwan from 2012 to 2020, (b) the monthly workload of shelter veterinarians from 2018 to 2020, and (c) factors associated with these trends.

Estimates reported in the multivariable regression are in terms of OR? If so, interpretation should consider the 95% CI, please clarify. Also, multivariable model results should be part of the manuscript and not in the Supplementary_files.

Minor comments/suggestions are highlighted in the attached .pdf file.

Author Response

Reviewer 3

Authors aims to investigate the trends in (a) animal intakes and outcomes of public shelters in Taiwan from 2012 to 2020, (b) the monthly workload of shelter veterinarians from 2018 to 2020, and (c) factors associated with these trends.

Estimates reported in the multivariable regression are in terms of OR? If so, interpretation should consider the 95% CI, please clarify. Also, multivariable model results should be part of the manuscript and not in the Supplementary_files.

Minor comments/suggestions are highlighted in the attached .pdf file.

Author response: Thank you for your time and suggestion. We are very grateful for your comments on the manuscript. We used multivariable linear regression instead of logistic regression, and the detailed p-value was in supplementary files. We have supplemented Table 4. to show the most important covariates in the final multivariable linear regression models. We haven’t added the Table of multivariable linear regression models in the manuscript because the indicators were transformed for linear regression and the results aren’t intuitive for the readers to get information from the Table. We have supplemented our Methods as shown below. According to your advice, we have amended the relevant part of the manuscript by putting (95% CI) after the % mentioned in the attached .pdf.

First, we transformed the data that violated the assumption of normality. After that, a Ridge regression model was built for each indicator using all covariates. Covariates with a coefficient larger than 0.01 were chosen for inclusion in the following model selection using the Akaike information criterion (AIC) and Bayesian Information Criterion (BIC). The best subset of variables from all possible combinations of the covariates (i.e. the final model) for each indicator was the one with the lowest weighted sum of standardised differences between AIC and BIC (Buendia et al., 2021). (Lines 213-220)